# Classifying Brain Tumors on Magnetic Resonance Imaging by Using Convolutional Neural Networks

**Marco Antonio Gómez-Guzmán** [1] , **Laura Jiménez-Beristaín** [2] , **Enrique Efren García-Guerrero** [1] , **Oscar Roberto López-Bonilla** [1] , **Ulises Jesús Tamayo-Perez** [1] , **José Jaime Esqueda-Elizondo** [2] , **Kenia Palomino-Vizcaino** [2] and **Everardo Inzunza-González** [1,*]

[1] Facultad de Ingeniería, Arquitectura y Diseño, Universidad Autónoma de Baja California, Carretera Transpeninsular Ensenada-Tijuana No. 3917, Ensenada C.P. 22860, Baja California, Mexico
[2] Facultad de Ciencias Químicas e Ingeniería, Universidad Autónoma de Baja California, Calzada Universidad No. 14418, Parque Industrial Internacional, Tijuana C.P. 22390, Baja California, Mexico
* Correspondence: einzunza@uabc.edu.mx; Tel.: +52-646-152-8244

**Abstract:** The study of neuroimaging is a very important tool in the diagnosis of central nervous system tumors. This paper presents the evaluation of seven deep convolutional neural network (CNN) models for the task of brain tumor classification. A generic CNN model is implemented and six pre-trained CNN models are studied. For this proposal, the dataset utilized in this paper is Msoud, which includes Fighshare, SARTAJ, and Br35H datasets, containing 7023 MRI images. The magnetic resonance imaging (MRI) in the dataset belongs to four classes, three brain tumors, including Glioma, Meningioma, and Pituitary, and one class of healthy brains. The models are trained with input MRI images with several preprocessing strategies applied in this paper. The CNN models evaluated are Generic CNN, ResNet50, InceptionV3, InceptionResNetV2, Xception, MobileNetV2, and EfficientNetB0. In the comparison of all CNN models, including a generic CNN and six pre-trained models, the best CNN model for this dataset was InceptionV3, which obtained an average Accuracy of 97.12%. The development of these techniques could help clinicians specializing in the early detection of brain tumors.

**Keywords:** brain tumor; neuroimaging; image classification; deep-learning; transfer-learning; CNN; MRI; artificial intelligence

## 1. Introduction

A brain tumor (often abbreviated as BT) is an abnormal growth of brain cells that may manifest symptoms of cancer [1]. It forms an abnormal segment with varying features of normal cells [2,3]. Benign BTs are composed uniformly of inactive cells. In contrast, malignant tumors are composed of cancerous, active cells with a non-uniform structure. These tumors can be divided into two categories: primary and metastatic BT. In the case of primary tumors, the cancerous cells are contained within the brain; however, in the case of metastatic tumors, the cancerous cells have spread to other parts of the body, not just the brain [4,5]. Because it affects people of all ages, BT is of the deadliest illnesses in the world [6]. It is the most prevalent malignancy in older persons and the third most prevalent among adolescents and young adults [4]. Brain tumors include Glioma, Meningioma, and Pituitary [7]. Gliomas are found in certain brain regions, such as the cerebral pedicle and spinal cord. They induce symptoms such as vomiting, headaches, and discomfort. They represent one-third of all brain tumors and 80% of primary malignant brain tumors [8]. Glioma cases are growing at an alarming rate with a serious impact on human mortality. Similarly, a Meningioma is a tumor that forms in the meninges, layers that surround the brain and spinal cord. A Pituitary tumor is an irregular enlargement that forms in the Pituitary gland. In most cases, these tumors are benign [9].

For tumor classification purposes, the term No-tumor is used in the literature to refer to the healthy brain category. According to the World Health Organization (WHO) Classification of Tumors of the Central Nervous System WHO 2021, BTs are divided into four classes (I-IV) with progressively higher malignancies and a worse prognosis [7]. In clinical practice, the kind, size, location, and grade of a tumor impact the selection of treatment [10–12]. In addition, medical imaging encompasses some noninvasive procedures for viewing the body's interior. They are an essential source of information for illness diagnosis nowadays. Therapy and diagnosis are the principal uses of medical imaging in the human body. Therefore, it contributes significantly to the improvement of human health [1,13,14]. Computed tomography (CT), X-ray, ultrasound imaging (UI), single photon emission computed tomography (SPECT), positron emission tomography (PET), positron magnetic resonance imaging (PET-MRI), magnetic resonance spectroscopy (MRS), and magnetic resonance imaging (MRI), which are used to diagnose BT, are some imaging techniques commonly used by specialists [15–17].

According to previous research, MRI is the most effective and extensively used method for identifying and classifying BTs. T1-weighted MRI (T1), T2-weighted MRI (T2), T1-weighted contrast-enhanced MRI (T1-CE), and fluid-attenuated inversion recovery (FLAIR) are the four diagnostic MRI modalities [16,18,19]. Brain tumor diagnosis using MRI employing software-based tools includes segmentation, identification, and categorization of brain tumors [5,20], which results in a quicker response to therapy and increases patient survival [16,21]. As a result, software specialists have been tasked with developing tumor detection systems, particularly using image processing [2]. Images, on the other hand, form a massive element of both digital and physical data stores. Because of this, image datasets tend to be relatively huge. The proliferation of digital cameras and other imaging technology has resulted in a huge uptick in the quantity of digital photographs taken and stored [14].

The manual identification and categorization of brain tumors in huge databases of medical images in typical clinical jobs have a significant cost in both effort and time. As a result, certain solutions have been adopted today, utilizing machine learning (ML) and deep learning (DL) approaches for brain tumor segmentation, detection, and classification [16]. DL approaches with CNN structures are now employed to analyze medical images of various forms of malignancy [22]. Similarly, the transfer learning (TL) technique has been utilized, which is defined as a process in which a model previously trained in a specific problem is used in another similar problem and which has the benefit of a shorter training period because it has already been trained with a similar problem. Both strategies yielded great results [23]. Sultan et al. [24], for instance, proposed a convolutional neural network (CNN)-based deep learning model for classifying three kinds of brain cancers using two publically accessible datasets. The total *Accuracy* of the suggested network structure was 96.13% and 98.7% for the two sets. Similarly, in Aamir et al., 2022 [17], the authors suggest two DL models for feature extraction from a dataset, to use yet a third model for the classification process. The strategy they developed was so effective that it led to a 98.95% success rate in classifying data. Additionally, Chattopadhyay and Maitra 2022 [13] created an algorithm for segmenting BT from MRI images by first employing a CNN and then utilizing conventional classifiers. It was found that the proposed model was 99.74% accurate.

On the other hand, Nayak et al., 2022 [25] showed a CNN using min-max normalization to classify 3260 MRI images into four categories (Glioma, Meningioma, Pituitary, and No-tumor). The developed network is a variant of EfficientNet. The results indicated that the model was 99.97% accurate during training and 98.78% during testing. Wahlang et al., 2022 [26] determined if an MRI image is normal or pathological. They created a DL architecture based on LeNet. Age and gender are now considered criteria as well. Compared to other pre-trained models, the LeNet-inspired model achieved an overall *Accuracy* of 88%, whereas CNN-DNN architectures achieved just 80%, support vector machine (SVM) 82%, and AlexNet (64%). Another work where the TL AlexNet

model was used as a tool in classification tasks was Badjie et al., 2022 [27]. They proposed a binary classification to identify between an unhealthy brain and a healthy brain. They obtained an *Accuracy* of 99.62%.

For Raza et al., 2022 [28], the categorization of three kinds of brain tumors (Glioma, Meningioma, and Pituitary) was accomplished using a hybrid deep-learning model named DeepTumorNet. Instead of these last five levels, they took the GoogLeNet design as a foundation and built 15 more layers. They achieved 99.67% *Precision*, 99.6% *Accuracy*, 100% *Recall*, and a 99.66% F1 score. They compared their outcomes to pre-trained models such as AlexNet, Resnet50, darknet-53, Shufflenet, GoogLeNet, SqueezeNet, Resnet101, Exception-Net, and MobileNetv2.

Another study point is using proposed and pre-trained TL models with other features or classification methods. An example of this is Maqsood et al. [29], who divided their method into five steps, including the design of a 17-layer deep neural network architecture for brain tumor segmentation, the use of modified MobileNetV2 CNN for feature extraction, and the application of M-SVM for the classification of Meningioma, Glioma, and Pituitary. They obtained a 97.47% *Accuracy* rate for the BraTS 2018 dataset and a 98.92% *Accuracy* rate for Figshare. Likewise, Amran et al., 2022 [30] established a deep hybrid learning classification model for binary brain tumors. This technique combines the GoogLeNet architecture with a CNN model by eliminating five levels of the GoogLeNet architecture and adding fourteen layers of the CNN model that automatically extracts features. ResNet, VGG-16, SqeezNet, AlexNet, MobileNetV2, and several ML/DL models were compared to the proposed model. Its classification scores were 99.51% *Accuracy*, 99% *Accuracy*, 98.90% *Recall*, and 98.50% F1-Score. Samme et al., (2002) [31], provided a deep hybrid transfer learning (GN AlexNet) model for Pituitary, Meningioma, and Glioma BT classification. The suggested model integrated the GoogleNet architecture with the AlexNet model by deleting five GoogleNet levels and adding ten AlexNet layers, which automatically collects and classifies characteristics. Comparing the proposed model against pre-trained models (VGG-16, AlexNet, SqeezNet, ResNet, and MobileNet V2). The model's *Accuracy* was 99.51%, and its sensitivity was 98.90%.

In other efforts, such as Ghazanfar et al., 2022 [32], a strategy for classifying Glioma tumors was created employing CNN for feature extraction and SVM for classification. They attained an *Accuracy* of 96.19% for the HGG Glioma type and 95.46% for the LGG Glioma tumor type when classifying the four Glioma types using the suggested method (edema, necrosis, enhancement, and non-enhancement). Jibon et al., 2022 [33] proposed a classification method to distinguish cancerous and non-cancerous tumors from MRIs using log-polar transform (LPT) and CNN. The LPT has been applied for the extraction of rotation and scaling features from distorted images, while the integration of CNN introduced an ML approach to the classification of tumors from distorted images. The results showed that the ML approach provides better classification, with a success rate of about 96%, on both single MRI images and brain MRI images with rotation and scale invariance. The model proposed by Yazdan et al., 2022 [34], is a multiclass classification solution for magnetic resonance imaging (MRI) in Glioma, Meningioma, Pituitary, and No-tumor. The experimental findings demonstrated that the suggested multi-scale CNN model outperforms AlexNet and ResNet in terms of *Accuracy* and efficiency while incurring less computational expense. The proposal has an *Accuracy* score of 91.2% and an F1 score of 91%. Ullah et al., 2022 [35] introduced a binary classification Tumor Resnet DL model for brain detection. The suggested model obtained 99.33% *Accuracy*. These experimental results, including the cross-dataset configuration, indicate that the TumorResNet model is better than some contemporary frameworks.

Another approach to TL was that of Alanazi et al., 2022 [36] who developed a TL-based model for classifying BT into its subtypes, including Pituitary, Meningioma, and Glioma. Using the notion of TL, they repurposed the isolated 22-layer CNN model of binary classification (tumor or No-tumor) to rescale the weights of neurons in order to classify MRIs. Consequently, the *Accuracy* of the constructed TL model for the employed images was

95.75%. Other cases of the use of TL are, for example, Ahmed et al., 2022 [37], who created an evolutionary method to identify MRI images into three brain tumor classifications. The Xception model was used for feature extraction. In this study, the model *Accuracy* was 99.06%. Secondly, Ullah et al., 2022 [38], utilized TL to compare nine classifiers on the same dataset. InceptionResNetv2, InceptionV3, Xception, ResNet18, ResNet50, Resnet101, ShuffleNet, DenseNet201 and MobileNetV2. The best model was InceptionResNetV2 and its *Accuracy* was 98.91%. Similarly, Deepak and Ameer 2019 [11] classified brain tumors into three BT classes. They utilized a pre-trained Google Neural Network to extract features from MRI. Moreover, SVM and K-nearest neighbors (KNN) were used. Their findings demonstrated an *Accuracy* of 98%.

In contrast, there are some works that only use TL models and assemble them for the classification task. For example, Kumar et al.[15] submitted a work that used the TL method. They focused on identifying malignant tumors, benign tumors, and healthy brain tissue. They made use of ResNet152. Using the CoV-19 OA optimization technique, the weight parameters were modified. They compared their findings to those of previously suggested models. The suggested approach achieved the *Accuracy* values of 99.57%, 97.28%, 94.31%, 95.48%, 96.38%, 98.41%, and 96.34%. With a different approach, the work by Tandel et al., 2021 [39], developed four therapeutically useful datasets. Five-fold cross-validation was used to evaluate the four sets with five DL-based models, AlexNet, VGG16, ResNet18, GoogleNet, and ResNet50, and five ML-based models, support vector machine, K-nearest neighbors, Nave Bayes, decision tree, and linear discrimination. They presented the MajVot method in order to maximize the classification performance of five DL and ML-based models. As a consequence, they achieved an increase in average *Accuracy*. Subsequently, using a majority voting technique, Tandel et al., 2022 [40] worked with five pre-trained CNNs, including AlexNet, VGG16, ResNet18, GoogleNet, and ResNet50, for three distinct datasets. Their maximum level of *Accuracy* was 99.06%. In a similar way, Wu et al., 2022 [41], using the TL models InceptionV3, Resnet101, and Densenet201, offered a similar categorization of three kinds of cancers. The average *Accuracy* for each model was 96.21%, 97%, and 96.5%, respectively.

The approach developed by Kazemi et al., 2022 [14] is a parallel CNN model comprised of AlexNet and VGGNet networks. The layer structure differs between the two network types. The characteristics are integrated at a single location, followed by the classification using the softmax function. The *Accuracy* of the model was 99.14% for binary classification and 98.78% for multiclass.

Five pre-trained models and one suggested CNN model were proposed by Aurna et al., 2022 [23], who picked the best models to combine them. VGG19, EfficientNetB0, InceptionV3, ResNet50, Xception, and the authors' suggested model were the CNNs employed in this work. They used three distinct datasets and reached an *Accuracy* of 99.67%, 98.16%, and 99.76%, with the best assembly, respectively.

This paper proposes a CNN-based magnetic resonance imaging classification approach for four classes of BT: Meningioma, Glioma, Pituitary tumor, and No-tumor, using the Generic CNN model and other pre-trained models. The used dataset, Msoud Msoud2021, is a combination of three datasets, Fighshare, SARTAJ, and BR35H, which includes 7023 images divided into 80% for the training and 20% for the test stages. For training the proposed DL models, the k-fold cross-validation approach is adopted. The procedure becomes more difficult by adding zoom and brightness in the preprocessing stage. Moreover, a novel tool called WandB [42], recently created by Weights & Biases for graphing and visualization of machine learning results is used. This study was motivated by the success of previous research in the TL field. To achieve the classification of BT in the target dataset, we applied different TL models previously described in the literature, such as InceptionResNetV2, InceptionV3, Xception, ResNet50, MobileNetV2, and efficientNetB0. In addition, the proposed CNN models are compared with various techniques to establish their effectiveness in MRI classification, one of them being a Model Size versus Model *Accuracy* graph, among other performance metrics. The goal is to discover the optimal

classifier for this quadruple class of brain tumors, which is a multiclass problem, and the complexity is higher.

The remaining sections of the article are structured as follows: Section 2 provides the proposed approach to the CNN models. In addition, the six pre-trained models, the generic CNN model, and the dataset are described. Section 3 presents the findings and the limitations of the work. Finally, in Section 4, the conclusions of the offered research are presented.

## 2. Materials and Methods

### 2.1. Proposed Method

The Google Colab platform and the Python programming language in version 3.8.10 are utilized for coding the preprocessing techniques. Version 2.9.2 of Tensorflow is used. Google Colab platform uses the NVIDIA A100-SXM4-40 GB, a professional graphics card.

On the other hand, classification plays an important role, as it organizes images into specific groups. It is the initial step in predicting an area or region containing abnormalities in the diagnosis of any disease [26,43].

This section describes the proposed technique for multiclass classification of BT, based on the four-step fundamental workflow of a CNN-based BT classification study proposed in [10]. The proposed approach consists primarily of four stages. First, the research dataset is retrieved from the Kaggle database [44]. This dataset is a composite of three distinct datasets that will be explained in subsequent sections. The dataset is preprocessed using a variety of approaches, including resizing, labeling, and data augmentation (Rotation, Zoom or Scale, and Brightness). Subsequently, the training and validation techniques are carried out using the Generic CNN model as well as pre-trained CNN models. In Figure 1, we can observe the models used in this study in order to verify their performance using the same dataset for the proposed CNN models. Likewise, the test is performed to verify that the training had been conducted correctly. Then, the test is performed by generating a test dataset with different images from the dataset used for training and validation.

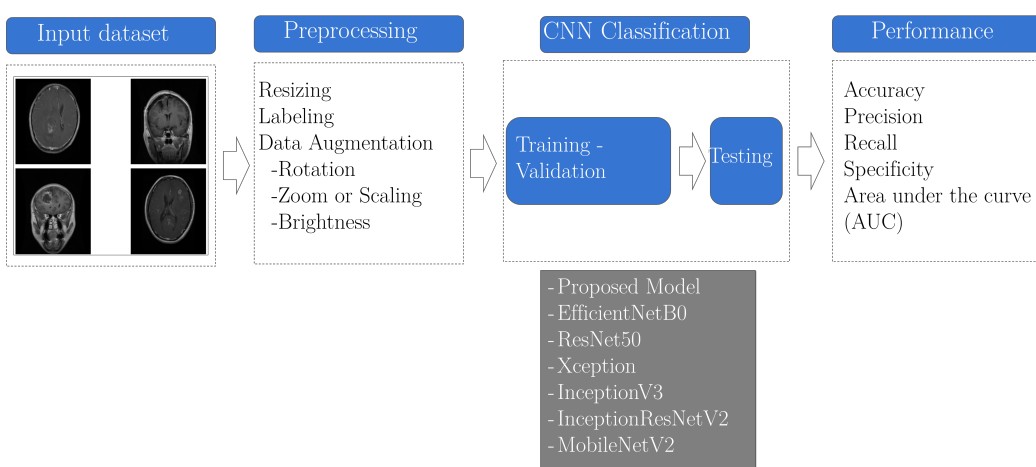

**Figure 1.** Flow diagram of the suggested method for BT classification.

Finally, the performance of the proposed models is assessed using the primary performance metrics such as *Accuracy*, *Precision*, *Recall*, *Specificity*, and area under the curve (AUC). The performance is compared to the pre-trained models, and the proportion of losses during training is also verified. All of those mentioned above, in order to determine which model performed better with the given dataset.

## 2.2. Preprocessing

For the coding of the preprocessing techniques, the Google Colab platform and the Python programming language in version 3.8.10 are used. Version 2.9.2 of Tensorflow is used. This platform uses The A100-SXM4-40 GB, a professional graphics card manufactured by NVIDIA.

This research proposes the following preprocessing procedures for the dataset: Resizing, Labeling, and Data Augmentation (Rotation, Zoom or Scaling, and Brightness), as depicted in Figure 1. In turn, the procedures of Rotation and Zoom or Scaling were applied as Position Augmentation methods. Brightness was also employed as a Color Enhancement Technique. These preprocessing methods were discussed in Xie et al., 2022 [10].

The preprocessing techniques started by declaring in a dictionary using the Python function *dict()*. This dictionary has been given a name and may be used to produce a new dataset with the properties it contains. It is recommended that this grouping of arguments be performed at the beginning of this stage in order to prevent excessive code in the instructions of this stage.

As mentioned in [10], because deep neural networks need inputs of a specific size, it is required to scale all images prior to their input into CNN classification models. As a result, the first approach used was Resizing, which employs the *rescale* parameter, which refers to Python function parameters.

Following that, the Labeling procedure is carried out. This procedure is characterized by the application of labels to each class. The quantity of produced labels is dependent on the number of classes handled by each research. For this research, the names of the four classes (Glioma, Meningioma, No-tumor, and Pituitary) and their corresponding images were established and allocated.

On the other hand, Data Augmentation is one of the most essential data approaches for addressing the issues of imbalanced distribution and data sparsity [10]. Numerous research studies on the classification of brain tumors have used this method, which incorporates geometric transformation operations such as rotation, zoom or scale, and brightness. During this stage, the *tf.keras.preprocessing.image.ImageDataGenerator()* function was also used to generate images [45]. For the rotation parameter three functions are proposed. One of them is *rotation_range*. This option helps establish the range of degrees for random image rotations. Another parameter used to rotate the image is *horizontal_flip*. It flips randomly the inputs horizontally. Moreover, *shear_range* was used because it randomly modifies the shear angle in a counter-clockwise direction in degrees. The following *zoom_range* parameter was used for the Zoom or Scale approach, which randomly zooms in or out on each image in the dataset. Finally, the *brightness_range* parameter was utilized to adjust the level of brightness, which gives a change to the random brightness intensity for each of the images in the dataset.

Table 1 summarizes the proposed parameters for preprocessing the CNN models. For details and background about the preprocessing parameters, please refer to [10,45]. If you wish to use more than these parameters, please note that they may increase the complexity of the training.

**Table 1.** Parameter values for the preprocessing stage for each CNN model.

| Parameter | Value |
| --- | --- |
| Rescale | 1.0/255 |
| Rotation | 10 |
| Horizontal flip | True |
| Shear | 0.1 |
| Zoom/Scaling | 0.1 |
| Brightness range | 0 to 0.7 |
| Labeling | 4 Labels |

### 2.3. Generic CNN Model

In this study, we use a generic CNN design to classify MRI images into four classes (See Figure 2). The CNN architecture classifies each pixel of an MRI image (slice) using one of four output labels: Glioma−0, Meningioma−1, No-tumor−2, Pituitary−3. It consists of 17 layers, beginning with the input layer, which contains the augmented images from the preprocessing stage, continuing through the convolution layers and their ReLU activation functions, plus batch normalization, max-pooling, a dropout layer, and concluding with a dense layer with softmax activation function to predict the output.

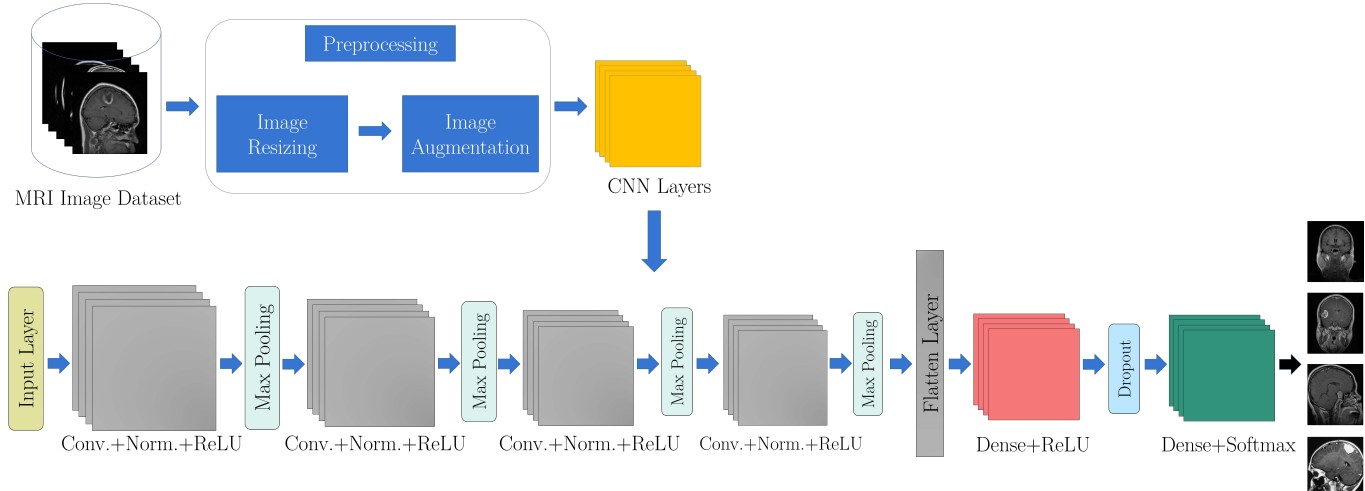

**Figure 2.** Architecture of the generic CNN model. Input size: 256 × 256. Model: four convolutional layers, with batch normalization, ReLu activation function, and max-pooling layers, applying at the end of the dense stage a Dropout of 0.5 and a dense layer with softmax activation leading to a classification into one of the four possible output labels.

The generic CNN model is simulated using Python programming language in Google Colaboratory. The parameter settings are given as follows: learning rate: 0.0001, dropout: 0.2, batch size: 16, epoch count: 10 per k-fold, optimizer: SGD and activation: ReLU and Softmax. Table 2 provides information about the hyperparameters for this model.

**Table 2.** Hyper-parameter values for training the Generic CNN model.

| Hyper-Parameter | Value |
|---|---|
| Activation function | ReLU |
| Initial learning rate | 0.0001 |
| Optimizer | Adam |
| Batch size | 16 |
| Dropout rate | 0.5 |
| Epochs | 10 |
| Train-test split | 70–30% |
| Output activation function | Softmax |

### 2.4. Pre-Trained CNN Models

In the pre-trained network approach, there are several publically accessible models; in this research, we consider ResNet50, MobileNet v2, Xception, InceptionV3, Incepton Res-NetV3, and EfficientNetB0 as case studies.

**ResNet-50**: Figure 3 depicts the block diagram of pre-tained ResNet50 architecture. The variants of ResNet placed first in the ImageNet-based ILSVRC 2015 ranking challenge. The architecture was described at [46]. They are now used in a variety of machine vision-related activities. Resnet101, ResNet50 and Resnet18 residual networks are 101 layers,

50 layers, and 18 layers deep, respectively [38]. Resnet101 shows more accurate results than Resnet18 and ResNet-50 due to the greater depth of the TL algorithm. The architecture uses an input size of 224 × 224 and can be modified to 256 × 256.

ResNets employ direct access connections to bypass several network levels; this omission compresses the network and accelerates learning. A relationship exists between the residual network and feature inference. ResNet also tackles the issue of declining *Precision*. Figure 4 illustrates the structure of the ResNet model.

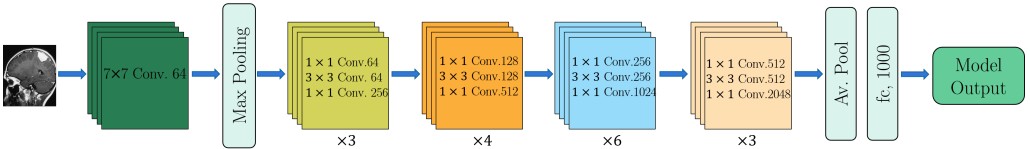

**Figure 3.** A basic block diagram representation of pre-trained ResNet50 architecture.

ResNet50 is utilized as a model for transfer learning in this study. Transfer learning is adaptable; the pre-trained model is utilized directly to identify images. The model is trained using the dataset examined for this study, and the fourth layer of neurons is added owing to the dataset's four classifications [30,47].

**Xception**: CNN Xception [23] utilizes point convolution and depth convolution. It is not essential to conduct convolution on all channels in this arrangement. This decreases connections and, hence, parameters. Like ResNet, it utilizes residual connections to enhance *Precision* [38]. There are input flow, middle flow, and exit flow (See Figure 4). This architecture's input size is 256 × 256. The Xception model's framework is readily modifiable. Because it has been trained using the ImageNet dataset, the pre-trained version of the Xception method can classify new jobs.

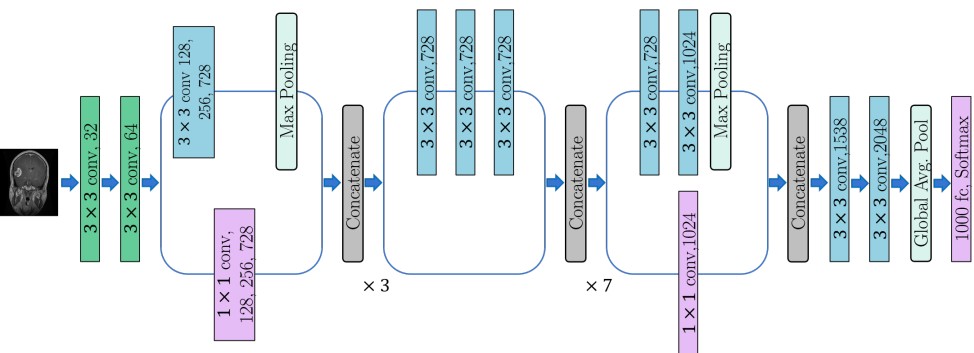

**Figure 4.** A basic representation of pre-trained Xception architecture.

**InceptionV3**: Inceptionv3 is a member of the Inception family of deep neural networks. It is a result of the enhancements made to the original Inception architecture. It is a deeper network owing to its arrangement of few connections. It is mostly composed of multiple Inception modules. Each module receives input from the preceding module. This architecture is 48 layers deep and has an input size of 299 × 299, but this can be modified [23,38]. The offered pre-trained version of Inceptionv3 is trained on the ImageNet database and can classify images of one thousand distinct items. Figure 5 depicts its architecture in its simplest form.

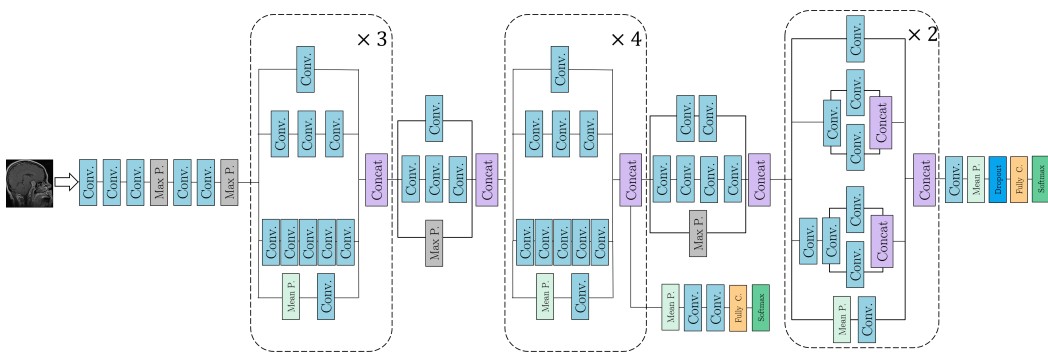

**Figure 5.** A basic representation of pre-trained InceptionV3 architecture.

**EfficientNetB0**: Tan and Le (2019) [48] introduced the EfficientNet network for scaling by balancing network height, depth, and input resolution to achieve greater *Precision*. The network uses an input size of 224 × 224 and the modified network uses an input size of up to 256 × 256. The original architecture's base network is depicted in Figure 6.

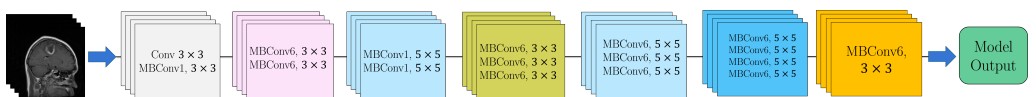

**Figure 6.** A basic representation of pre-trained EfficientNetB0 architecture.

**InceptionResNetV2**: InceptionResNetv2 [38] is a deep CNN built with Inception and a collection of residual connections. This network employs modified Inception blocks as opposed to the original Inception blocks and a filter expansion layer after each Inception block using convolution without activation. On top of the representative layers, batch normalization is used to enhance the number of Inception blocks. This network processes an image with 299 × 299 pixels as input.

**MobileNetV2**: The input image size for Mobilenetv2 is 224 × 224 pixels. Given its processing efficiency, this model is better suited for real-time and mobile applications. The Mobilenetv2 model's rapid execution is due in part to its use of point-wise and depth-wise convolution methods. The network makes use of the remaining links between the bottlenecks themselves. The Mobilenetv2 network consists of a 32-filter convolutional beginning layer, followed by 19-filter bottleneck layers.

Table 3 shows the hyperparameters utilized during training for the pre-trained TL models. These hyperparameters were chosen according to the state of the art [16,23,25,26,28–30,32,35–37].

**Table 3.** Hyper-parameter values for training the TL models.

| Hyper-Parameter | Value |
| --- | --- |
| Activation function | ReLU |
| Initial learning rate | 0.0001 |
| Optimizer | SGD |
| Batch size | 16 |
| Dropout rate | 0.3–0.4 |
| Epochs | 10 |
| Train-test split | 70–30% |
| Output activation function | Softmax |

### 2.5. Brain Tumor Kaggle Dataset

The publicly accessible MRI dataset from the Kaggle repository was utilized for training, validation, and testing of the multiple TL-based techniques employed in this work,

as well as the model. The Brain Tumor MRI dataset Msoud [44] is a composite of the three publicly accessible datasets listed below:

1. Figshare [49]
2. SARTAJ [50]
3. Br35H [51]

The details associated with the dataset are displayed in Table 4. This dataset contains 7023 MRIs of the human brain of different types in grayscale and JPG format. The four classes of brain tumors shown in the dataset are Glioma (with 1321 images for training and 300 for testing), Meningioma (with 1339 images for training and 306 for testing), No-tumor (with 1595 images for training and 405 for testing) and Pituitary (1457 images for training and 300 for testing). For the training and validation task, 80% and 20% of the images were used, respectively. However, in the preprocessing stage, Resizing and Data Augmentation were applied to the dataset to provide an adequate input size for each different model and increase the number of images to be used. This resulted in 9139 images, of which 70% of the dataset was used for training and 30% was used for testing.

**Table 4.** Dataset details.

| Classes | Images for Training | Images for Testing |
|---|---|---|
| Glioma | 1321 | 300 |
| Meningioma | 1339 | 306 |
| No-tumor | 1595 | 405 |
| Pituitary | 1457 | 300 |
| Total | 5712 | 1311 |

The images of the no-tumor class were extracted from the Br35H dataset. The Glioma class images in the SARTAJ dataset are incorrectly classified. Hence, they have been replaced by images from the dataset in the reference [49]. Figure 7 depicts some instances of the images stored inside the database:

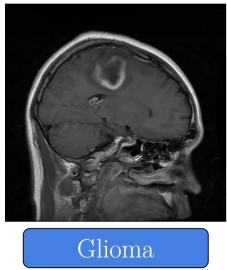 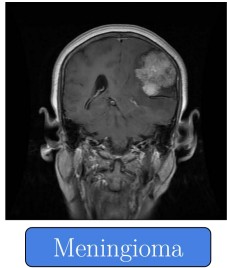 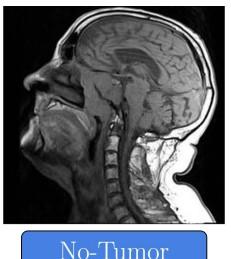 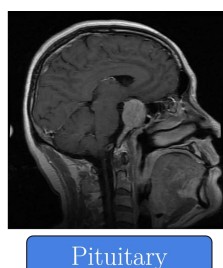

Glioma   Meningioma   No-Tumor   Pituitary

**Figure 7.** Sample of the MRIs in the dataset.

*2.6. Performance Metrics*

*Accuracy* in classification is the proportion of correct predictions and the total data elements [52], it is calculated according to Equation (1):

$$Accuracy = \frac{TP + TN}{TP + FP + TN + FN} \tag{1}$$

*Precision* is the proportion of optimistic forecasts belonging to the all positive category [52]. The equation of the *Precision* value is (2):

$$Precision = \frac{TP}{TP + FP} \tag{2}$$

The

*Recall*

is calculated by dividing the number of true positive (TP) outcomes by the total number of positive class elements [52]. The equation of *Recall* is (3):

$$Recall = \frac{TP}{TP + FN} \qquad (3)$$

Likewise, the *Specificity* described by Equation (4):

$$Specificity = \frac{TN}{(FP + TN)} \qquad (4)$$

Finally, we will display the proportion of losses as well as the AUC score, which reflects the model's capacity to differentiate between distinct classes, with a larger number indicating better performance [23,52].

For the equations above: True Positive (TP) is the number of anticipated positive cases that are, in fact, positive. True Negative (TN) is the number of anticipated negative cases that are in fact negative. False Negative (FN) is the number of expected negative situations that are in fact positive; it is also known as (type two) error. False Positive (FP) is the number of expected positive cases that are in fact negative; it is also known as (type one) error.

## 3. Results and Discussion

As stated in earlier parts, the neural network models were trained using a 5-fold cross-validation. The derived quantitative results are depicted in the Figures and Tables that follow. This part also evaluates the performance of different pre-trained TL classifiers used to classify MRI images from the multiclass dataset described in earlier sections. The primary benefit of TL classifiers and hyperparameter adjustment is the elimination of overfitting issues, which are common in DL algorithms when experimenting with a relatively small sample [38].

After the preprocessing task, the following image examples were obtained as shown in Figure 8. As can be observed, zoom and brightness characteristics were added to each image sample for each tumor type, as in the cases of Glioma and Meningioma tumors and the position were altered, as in the case of the Pituitary without the tumor. In the case of certain images, such as the Pituitary example, just the position and brightness were altered. The primary objective of the used preprocessing approaches was to enable the models to acquire knowledge independently of image features.

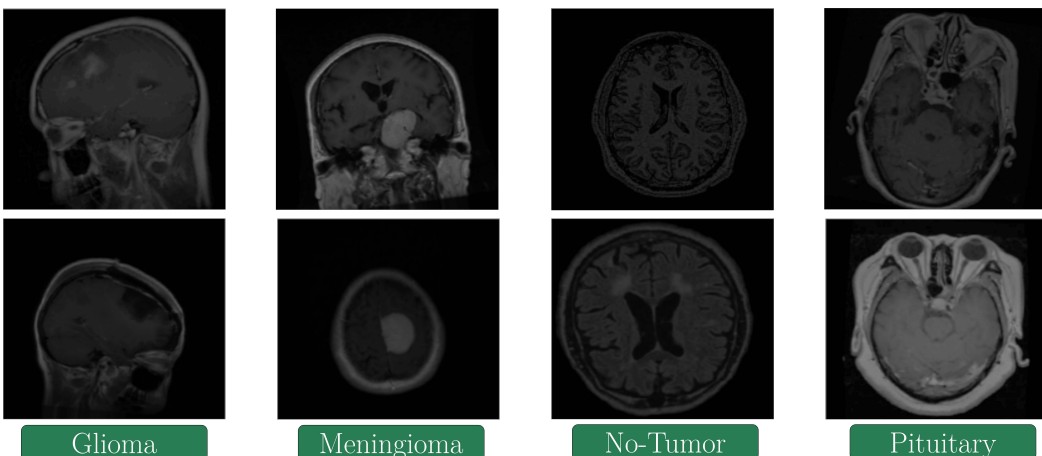

| Glioma | Meningioma | No-Tumor | Pituitary |

**Figure 8.** Examples of preprocessed images.

Table 5 shows the classification results of the TL algorithms and the proposed model, revealing that each TL classifier produced acceptable and competitive results, considering the preprocessing used and the composition of the dataset used. Likewise, it is observed that the generic CNN model does not have acceptable results compared to the pre-trained mod-

els, with an *Accuracy* of 81.05%. Using the evaluation parameters of *Accuracy*, *Precision*, *Recall*, *Specificity*, and area under the curve (AUC) the algorithms were evaluated. The loss ratio for each instance was also mentioned, noting that the InceptionV3 model was the model that obtained the lowest losses during the training stage. According to the results, the DL InceptionV3 model had the highest average *Accuracy* of 97.12%, the average obtained from the cross-validation performed with k = 5. On the other hand, it is essential to note that the variations of the ResNet, ResNet50, and InceptionResNetV2 produced different but extremely similar results, being the second and third place in the table with 96.97% and 96.78%, respectively.

**Table 5.** Performance Metrics results .

| Model | Accuracy | Precision | Recall | Specificity | AUC | Loss |
|---|---|---|---|---|---|---|
| InceptionV3 | 0.9712 | 0.9797 | 0.9659 | 0.9998 | 0.9984 | 0.0796 |
| ResNet50 | 0.9697 | 0.9796 | 0.9637 | 0.9997 | 0.9982 | 0.0812 |
| InceptionResNetV2 | 0.9678 | 0.9767 | 0.9623 | 0.9998 | 0.9980 | 0.0907 |
| Xception | 0.9567 | 0.9662 | 0.9509 | 0.9998 | 0.9972 | 0.1157 |
| MobileNetV2 | 0.9545 | 0.9661 | 0.9473 | 0.9997 | 0.9967 | 0.1221 |
| EfficientNetB0 | 0.9088 | 0.9312 | 0.8912 | 0.9997 | 0.9898 | 0.2347 |
| Generic CNN | 0.8108 | 0.8527 | 0.7677 | 0.9924 | 0.9616 | 0.4661 |

Table 6 shows in more detail the results obtained in each of the k-folds in the cross-validation process for the model that obtained the best average performance, InceptionV3. As can be observed, the standard deviation and the confidence limit have minor differences between each of the K-folds, which is a positive result since it demonstrates the distance between the data and the data median. This demonstrates the algorithm's robustness to diverse data blocks.

**Table 6.** Performance Metrics results of InceptionV3.

| K-Fold | Accuracy | Precision | Recall | Specificity | AUC | Loss |
|---|---|---|---|---|---|---|
| 1 | 0.9717 | 0.9788 | 0.9666 | 0.9998 | 0.9985 | 0.07778 |
| 2 | 0.9727 | 0.9818 | 0.9671 | 0.9999 | 0.9987 | 0.07591 |
| 3 | 0.9714 | 0.9801 | 0.9667 | 0.9998 | 0.9983 | 0.08086 |
| 4 | 0.9702 | 0.9788 | 0.9646 | 0.9998 | 0.9984 | 0.08082 |
| 5 | 0.9703 | 0.9791 | 0.9647 | 0.9999 | 0.9985 | 0.08284 |
| Average | 0.97126 | 0.9797 | 0.9659 | 0.9998 | 0.9984 | 0.0796 |
| Std. Desv. | 0.00093 | 0.00114 | 0.00106 | 0.00004 | 0.000132 | 0.00246 |
| Conf. Limit | 0.0008 | 0.001 | 0.0009 | 0.00004 | 0.0001 | 0.00216 |

Using Wandb, plots were generated. Wandb is a tool for tracking machine learning experiments. It makes it easy for machine learning practitioners to keep track of experiments and share their results with partners [42].

The training and validation process of the best classification model for this study is shown in Figures 9 and 10. The metrics shown are the average training and validation accuracy and the average of training and validation *Precision*. The number of k-folds used in this study may be seen in the graphs. The average percentages of *Accuracy* and *Precision* are 97.12% and 97.97%, respectively. On the other hand, the average validation *Accuracy* and *Precision* are 97.82% and 98.64%, respectively.

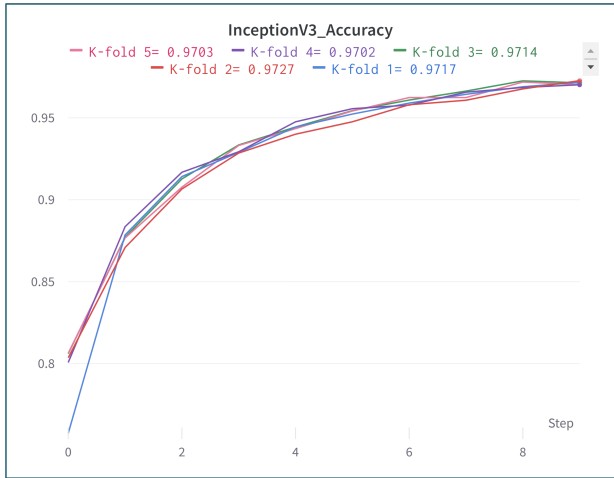
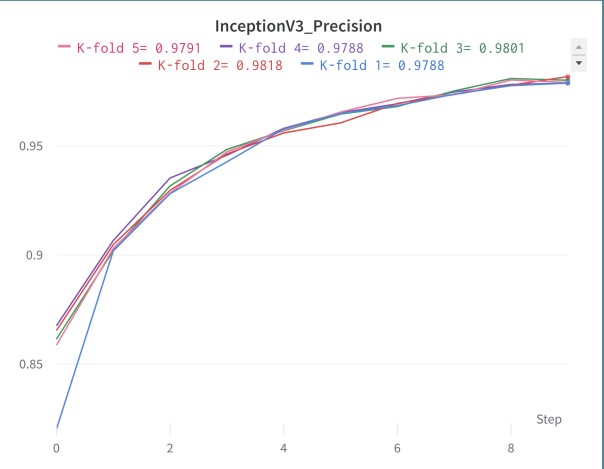

**Figure 9.** Training *Accuracy* and precision of the InceptionV3 model.

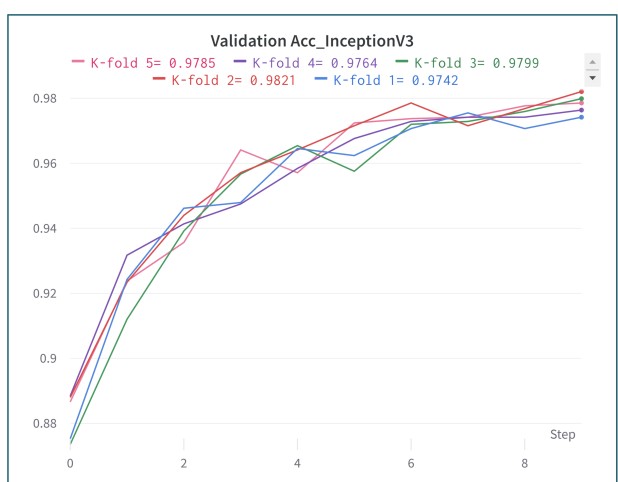
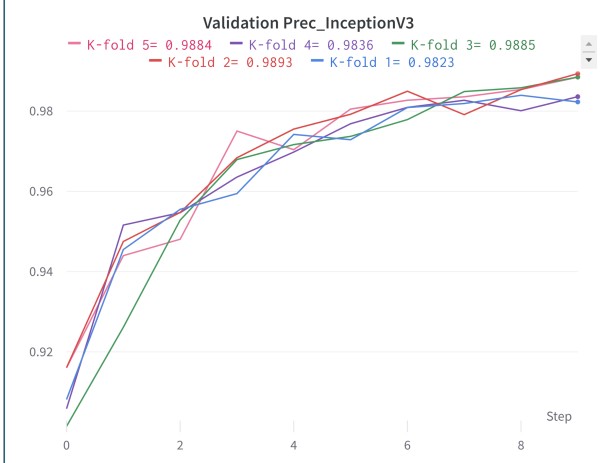

**Figure 10.** Validation *Accuracy* and *Precision* of the InceptionV3 model.

Moreover, Figure 11 shows the percentage of losses generated in every K-fold in the stage of training and validation of the InceptionV3 model. The average percentage of losses observed during training and validation of this model is 7.9% and 6.3%, respectively.

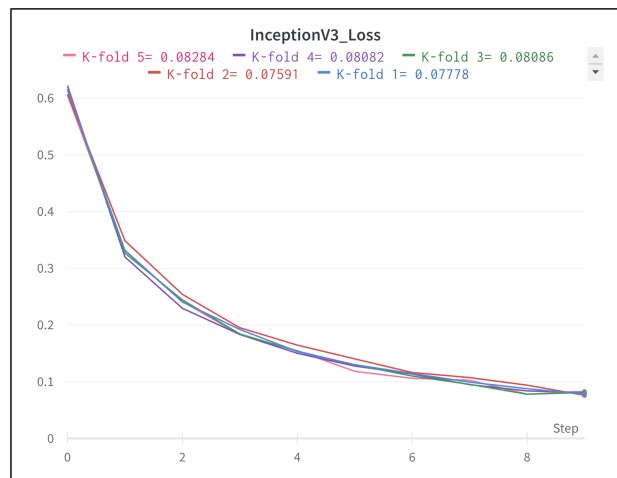
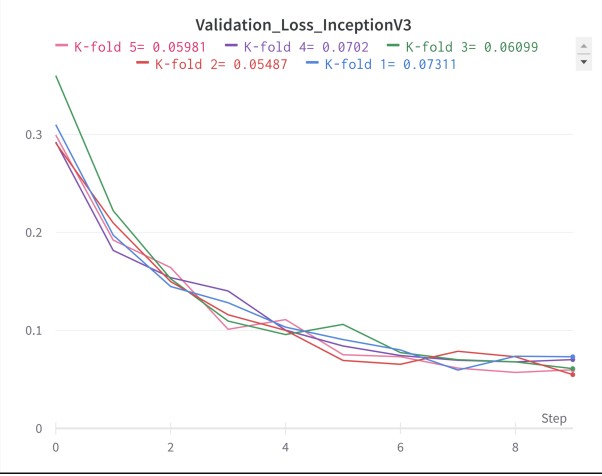

**Figure 11.** Training loss and validation loss of the InceptionV3 model.

Different performance metrics, such as *Accuracy*, *Precision*, and *Recall*, listed in Table 5, were used to compare the suggested model's performance. Using the confusion matrix, these parameters are examined. Figure 12 represents the confusion matrices used to study the specifics of each k-fold. As a result of overfitting using 30% of the test data extracted from the data set during the Data Augmentation phase, these confusion matrices include some misclassifications in each k-fold. The misclassified tumors of the InceptionV3 model in the confusion matrix of the first k-fold include 13 of label 0 corresponding to Glioma, 12 of label 1 corresponding to Meningioma, 26 of label No-tumor, and 4 of label Pituitary, as shown in Figure 12. Five from the Glioma label, thirty-four from the Meningioma label, nineteen from the No-tumor label, and one from the Pituitary label are misclassified tumors in the second k-fold confusion matrix. For k-fold equal to 3, 7 Gliomas, 16 Meningiomas, 12 No-tumor labels, and 1 Pituitary label are misclassified as tumors. For k-fold equal to 4, the misclassified tumors consist of one Glioma, 17 Meningiomas, 21 No-tumor labels, and 7 Pituitary labels. The final k-fold value is 5, and the misclassified classes consist of 5 Gliomas, 21 Meningiomas, 34 No-tumor labels, and 8 Pituitary labels. Due to less misclassified data, the InceptionV3 model is more accurate than the alternatives. k-fold classification of Glioma and Pituitary tumor is performed very effectively by any CNN model. The meningione and No-tumor classes cannot be learned as efficiently as the other three.

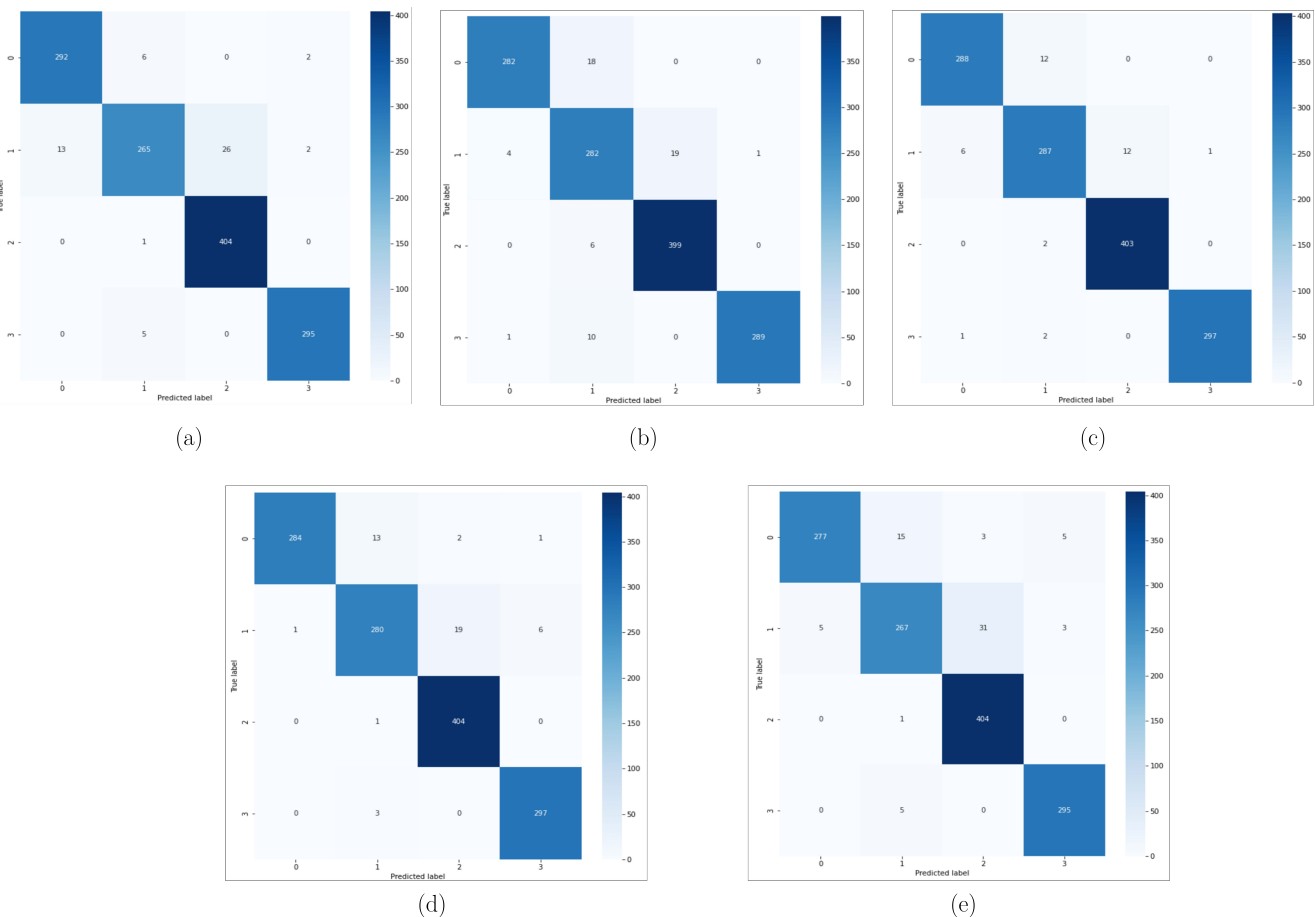

**Figure 12.** Confusion matrix in test stage of InceptionV3 model: (**a**) Confusion Matrix k-fold = 1; (**b**) Confusion Matrix k-fold = 2; (**c**) Confusion Matrix k-fold = 3; (**d**) Confusion Matrix k-fold = 4; (**e**) Confusion Matrix k-fold = 5.

Another method that helped us to determine the most efficient model in this study was the use of the Model Size versus Model *Accuracy* diagram. This is shown in Figure 13. This graphic is often used to evaluate the performance of a model with its size since the

latter determines the possible computing cost (this issue will be addressed in the next section). This kind of graphic is used in studies such as Tan and Le (2019) [48] where different pre-trained models and models proposed by the authors were compared. The graphic is constructed by placing the number of parameters or model size on the x-axis and the percentage of *Accuracy* on the y-axis. The sizes are listed in increasing order, beginning with the lowest model size regardless of its *Precision* and progressing to the biggest model size. The relative *Accuracy* of each model is then determined, and the graph's trajectory is defined by lines. Finally, the graph's structure is evaluated to determine which model is ideal in terms of parameters and *Precision*.

In this case study, InceptionV3 remains the most remarkable model. Although our perspective shifted with regard to ResNet50, which scored second in Table 5 since its parameters are greater than those of InceptionV3, which suggests a greater computing cost. Considering its *Precision* and small number of parameters, MobileNetV2 might be considered the second-best model for the dataset utilized in this investigation.

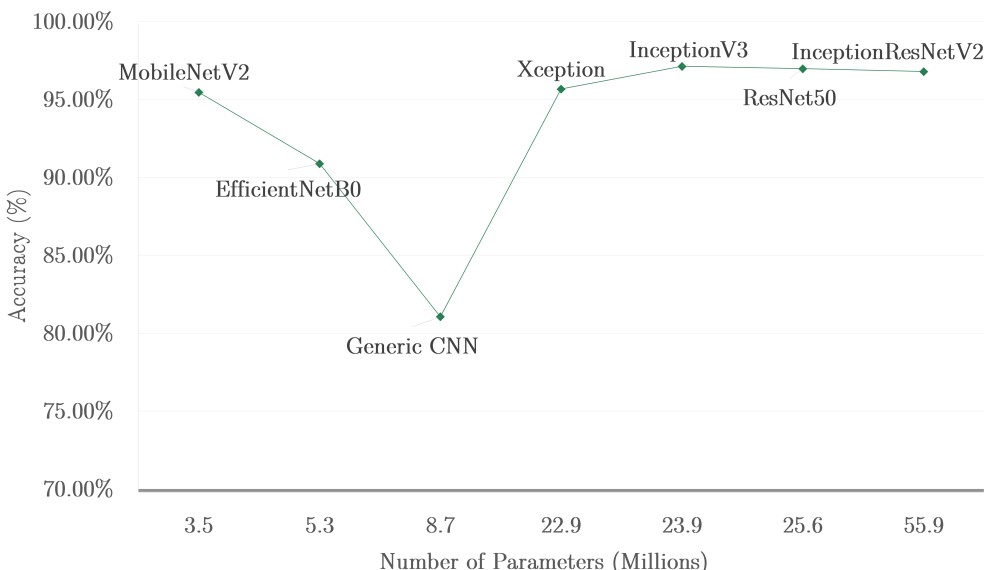

**Figure 13.** Model Size versus Model *Accuracy*. All numbers are indicated in millions for every single model.

### 3.1. Computational Complexity

This section discusses the computational complexity necessary for this research. Table 7 has seven columns, beginning with the name of the model, followed by the *Accuracy* and parameters of each model. Then, the computational cost is shown by percentage, beginning with the CPU and GPU usage, followed by the percentage of CPU memory allocated and concluding with the training time due to its relation with the computational cost. The latter will be presented in a bar chart and explained subsequently.

According to Table 7, the InceptionV3 model distinguishes itself from other high-*Accuracy* models because of its low GPU consumption and high *Accuracy*. Regarding the proportion of GPU Memory allocated, the InceptionV3 model achieved an average value of 60%. In terms of CPU utilization, the number for the InceptionV3 model is near the mean. As indicated in Table 7, the runtime of this model represents the greatest possibility for optimization.

For this study, the Google Colab platform was used. This platform used The A100-SXM4-40 GB, a professional graphics card manufactured by NVIDIA.

**Table 7.** Computational complexity of the models trained in this study.

| Model | *Accuracy* (Average) | Parameters (Millions) | % CPU Utilization | % GPU Utilization | % GPU Memory Allocated | Training Time (Minutes) |
|---|---|---|---|---|---|---|
| InceptionV3 | 0.9712 | 23.9 | 73.7 | 58.73 | 60 | 323 |
| ResNet50 | 0.9697 | 25.6 | 71.36 | 78.53 | 66.83 | 223 |
| InceptionResNetV2 | 0.9678 | 55.9 | 89.94 | 92.29 | 60.05 | 294 |
| Xception | 0.9567 | 22.9 | 75.36 | 98.4 | 66.83 | 274 |
| MobileNetV2 | 0.9545 | 3.5 | 73.45 | 40.49 | 32.95 | 231 |
| EfficientNetB0 | 0.9088 | 5.3 | 73.72 | 60.51 | 32.95 | 239 |
| Generic CNN | 0.8108 | 8.7 | 73.76 | 35.04 | 33.05 | 200 |

For all the models considered in this study, the following computational cost plots were generated using the WandB tool previously mentioned in the reference [42]. As can be seen in Figure 14, shows the percentage of GPU and CPU utilization in each training k-fold for the InceptionV3 model. In each case, the highest utilization value obtained was taken and the average of these was obtained, which are the CPU 73.7% and GPU 58.73% utilization percentage values shown for InceptionV3 shown in Table 7.

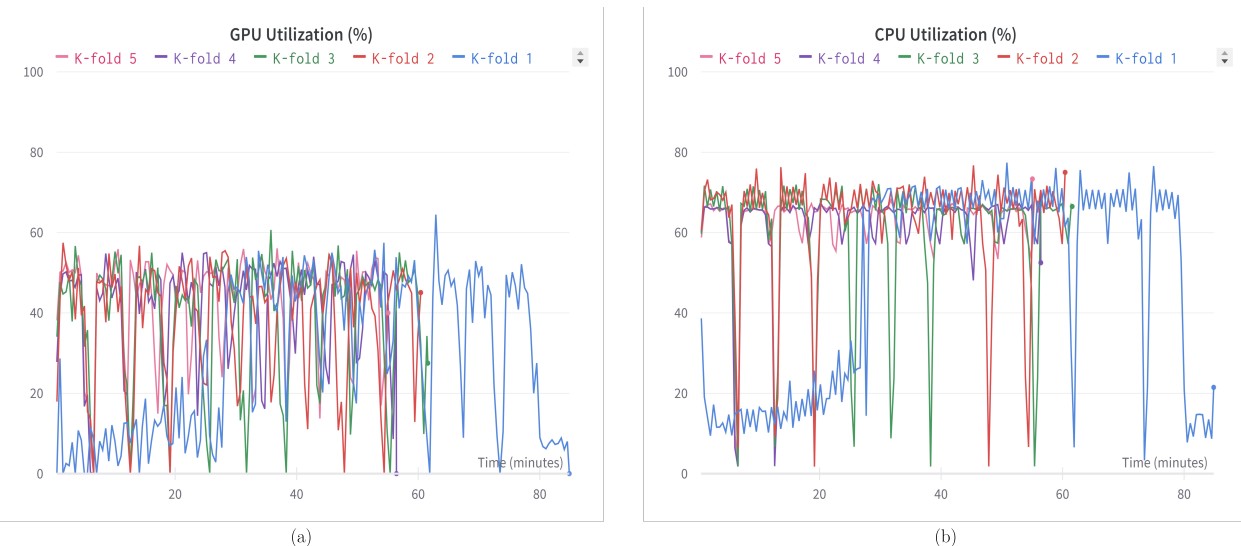

(a)

(b)

**Figure 14.** Computational resources used in InceptionV3 model: (**a**) GPU utilization percentage graph for InceptionV3 model; (**b**) CPU utilization percentage graph for InceptionV3 model.

Figure 15 additionally shows and demonstrates that 60% of GPU Memory was allocated while training k-folds.

As a result of the Data Augmentation approach, which greatly increased the size of the dataset, all classification models took a substantial amount of time. This time is also dependent on the model's complexity and architectural design. The duration is measured in hours and minutes. For brain tumor classification, TL's InceptionV3 model was the longest but most effective elapsed time classifier, delivering good classification results but consuming the greatest time (5 h and 23 min). ResNet50 required a maximum of 3 h and 43 min to detect and classify MRI images of brain tumors into distinct kinds. It was the TL model with the quickest execution time and the second-best in the results table. Due to its architecture, the MobileNetV2 model was also one of the models with the shortest run time, producing results in 3 h and 51 min with a lower computational cost than training with the EfficientNetB0 model, which took 3 h and 59 min, generic CNN model which took 3 h and 20 min. It should be noted that the classification time of the various Resnet

TL classifier versions grows as the number of framework layers increases. For instance, InceptionResNetV2 required a minimum of 4 h and 54 min. The length of the Xception models was 4 h and 34 min. Moreover, the training time of every CNN model was plotted in Figure 16. Table 5 demonstrates that, for the dataset utilized in this investigation, the *Accuracy* rose somewhat when a shallower ResNet model was employed. However, as the network's depth rises, so does its computational complexity and, consequently, its training time, which eventually impacts the network's efficiency; this explains why the *Accuracy* of the ResNet variations varies. In addition, we may deduce that the inceptionV3 algorithm is the best approach for classifying brain tumors in this study.

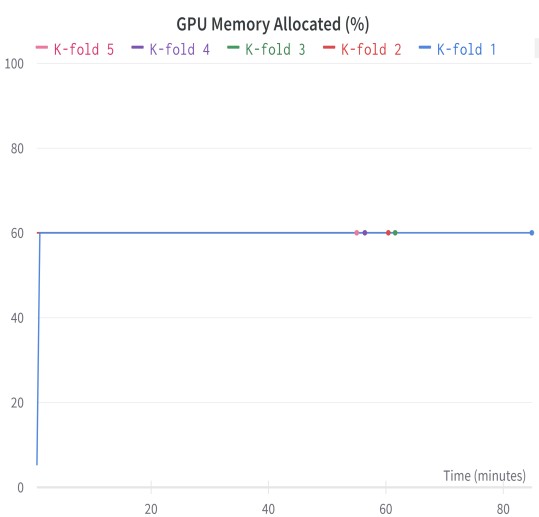

**Figure 15.** GPU Memory Allocated percentage graph for InceptionV3 model.

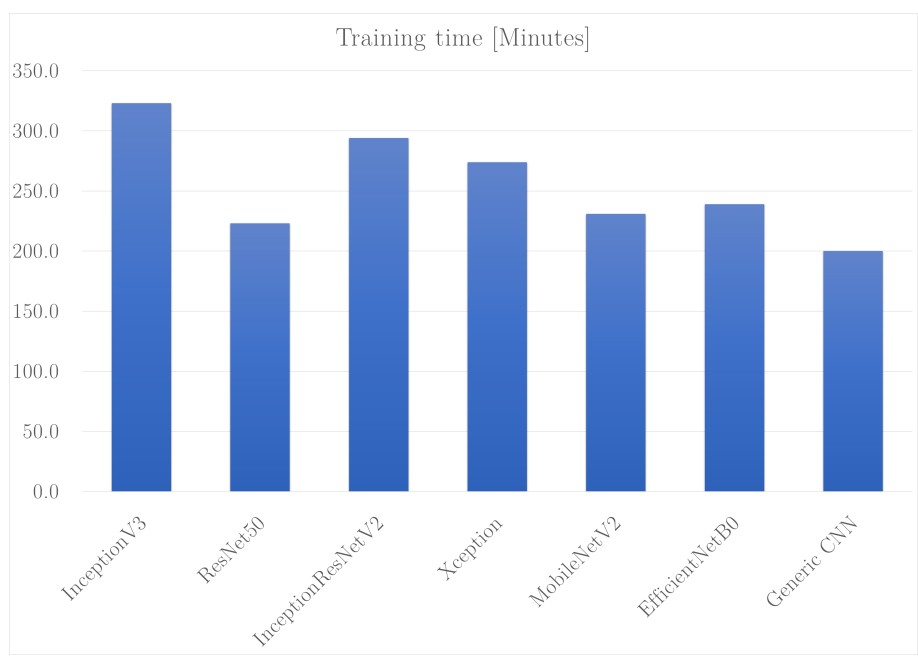

**Figure 16.** Training time (in minutes) of every model in this work.

### 3.2. Comparison to Contemporary Related Work

We evaluated the classification performance of the best deep neural network, namely InceptionV3, to existing approaches for categorizing brain cancers into four categories: Meningioma, Pituitary, Glioma, and No-tumor. Specifically, we compared the proposed

work to existing DL methods. In some of them, TL is the primary method, while in others, proposed models are the primary method.

Table 8 provides a detailed comparison of the pre-trained classification model techniques for brain tumor classification. Table 8 only includes *Accuracy* as the primary performance statistic since it is the metric most often employed in all relevant research. In the first column of Table 8 is the citation of the reviewed work, and the second column is the technique employed in each research, which may be an author-proposed model or the usage of pre-trained models. The third column additionally includes the dataset utilized in each research study since it is essential to refer to the dataset as a key indication of the results and to emphasize that in the current study, three distinct datasets, including a substantial number of photos are combined. In some research, the classification task is conducted, but the healthy brain is not taken into consideration as a class, an activity that is carried out in this study. In order to compare and assess the best current approaches, the best method or result attained is then included in each study.

**Table 8.** Comparison with related work.

| Reference | Model | Dataset | Classes | Best Model | *Accuracy* |
|---|---|---|---|---|---|
| Ref. [16] | CNN Multi Scale | Nanfang Hospital | 3 | - | 0.9730 |
| Ref. [24] | CNN | REMBRANDT | 3 | - | 0.9613 |
| Ref. [38] | TL | SARTAJ | 3 | InceptionResNetV2 | 0.9891 |
| Ref. [53] | CNN and SVM | Figshare | 3 | - | 0.9582 |
| Ref. [25] | Dense Efficient-Net | Figshare | 3 | Dense EfficientNet | 0.9997 |
| Ref. [36] | TL-CNN | Figshare, SARTAJ, BR35H | 3 | Developed TL CNN | 0.9575 |
| Ref. [26] | LeNet Inspired Model | Figshare, Brainweb, Radiopedia combination | 3 | - | 0.8800 |
| Ref. [28] | TL and DeepTumorNet | Figshare | 3 | DeepTumorNet | 0.9967 |
| Ref. [29] | Hybrid MobileNetV2 and TL | Figshare and BraTS 2018 | 3 | Hybrid MobileNetV2 | 0.9892 |
| Ref. [30] | Hybrid GoogLeNet and TL | BR35H | 2 | Hybrid GoogLeNet | 0.9910 |
| Ref. [35] | TumorResNet and TL | BTD-MRI dataset | 2 | Proposed TumorResNet | 0.9933 |
| This work | Generic CNN and six TL models | Figshare, BR35H and SARTAJ combination | 4 | InceptionV3 | 0.9712 |

According to Table 8, the InceptionV3 TL model outperforms some existing state-of-the-art methods. Due to its capacity to extract stronger and more distinguishable deep features for classification, the method yields the best outcomes. Moreover, although we do not employ a balanced dataset (brain tumor classification (MRI) dataset), the number of images is adequate for the network to train. In contrast, the datasets used in previous approaches, such as [30,35] described in the preceding table, only have two classes for classification and do not have a substantial number of photos. Therefore, the dataset utilized in this study exceeds the state-of-the-art in the number of images and classes. In some instances, the fifth column of Table 8 specifies the optimal model. Not in every study did the TL perform better than the recommended models. However, several InceptionV3 and InceptionResNetV2 models were among the best in certain trials. Similarly, several models were shown to be superior to the TL models.

### 3.3. Limitations of the Study

Finding an acceptable dataset for the classification problem is one of the constraints of this study. This kind of up-to-date medical information is difficult to collect, but fortunately, there are databases such as Kaggle that handle the storing of research data. In contrast, there is computational complexity. The usage of cloud-based technologies, such as Google Colab's free version, may be beneficial for completing such activities. Although it is feasible to obtain a pro version and optimize time and resources, a high-performance computer would be required to do this research.

### 4. Conclusions

This study compared six pre-trained models and one developed generic CNN model for classifying brain tumors. The objective of this work was to automate the classification of brain tumors by identifying the optimal DL classifier. Using a brain tumor MRI dataset, we evaluated seven CNN models, including InceptionResNetv2, Inceptionv3, Xception, Resnet-50, efficientnetB0, Mobilenetv2, and one generic CNN model. The identified brain tumors were Glioma, Meningioma, Pituitary, and No-tumor. The experimental results demonstrate that the pre-trained TL model InceptionV3 achieved the best *Accuracy* for classifying brain tumors in the previously described dataset. The *Accuracy* of 97.12% supports the validity of this model for classifying the four classes contained in the dataset. The classification *Precision* of 97.97% for BT demonstrates that the best model is better than some previous hybrid techniques for BT classification. Although the TL of six DL models for classification of brain cancers by MRI has been investigated in this study, more models need to be investigated.

A model size versus model *Accuracy* graph was also presented, which confirmed the effectiveness of the model chosen as the best for the study, due to the relationship between its parameters, the computational cost involved and the *Accuracy* obtained.

Despite the limited number of images and unbalanced classes in our dataset, we have obtained positive results. Using data augmentation techniques, we have increased the size of the training dataset and added additional features that facilitate model learning. The development of these techniques could help clinicians specializing in the early detection of brain tumors.

**Author Contributions:** Conceptualization, L.J.-B. and E.I.-G.; Data curation, U.J.T.-P. and K.P.-V.; Formal analysis, E.E.G.-G., L.J.-B. and U.J.T.-P.; Funding acquisition, J.J.E.-E., O.R.L.-B. and U.J.T.-P.; Investigation, M.A.G.G. and O.R.L.-B.; Methodology, M.A.G.G., L.J.-B. and E.I.-G.; Project administration, O.R.L.-B.; Resources, J.J.E.-E. and O.R.L.-B.; Software, M.A.G.-G.; Supervision, E.I.-G. and L.J.-B.; Validation, K.P.-V. and E.E.G.-G.; Visualization, U.J.T.-P. and O.R.L.-B.; Writing—original draft, M.A.G.-G.; Writing—review and editing, L.J.-B., J.J.E.-E. and E.I.-G. All authors have read and agreed to the published version of the manuscript.

**Funding:** This research had funds provided by the Universidad Autónoma de Baja California (UABC) through grants number 679, 300/2658 and 402/2575.

**Institutional Review Board Statement:** Not applicable.

**Informed Consent Statement:** Not applicable.

**Data Availability Statement:** The datasets utilized in this paper are Brain Tumor MRI dataset Msoud (https://www.kaggle.com/datasets/masoudnickparvar/brain-tumor-mri-dataset (accessed on 10 September, 2022)), Fighshare (https://figshare.com/articles/dataset/brain_tumor_dataset/1512427 (accessed on 10 October, 2022)), SARTAJ (https://www.kaggle.com/sartajbhuvaji/brain-tumor-classification-mri/metadata (accessed on 10 September, 2022)) and Br35H (https://www.kaggle.com/datasets/ahmedhamada0/brain-tumor-detection?select=no (accessed on 10 September, 2022)).

**Acknowledgments:** We want to thank UABC for all the support provided to the researchers. To CONACyT for the scholarship granted to M.A.G.-G.

**Conflicts of Interest:** The authors declare no conflicts of interest. The funders had no role in the design of the study; in the collection, analyses, or interpretation of data; in the writing of the manuscript, or in the decision to publish the results.

## Abbreviations

The following abbreviations are used in this manuscript:

| | |
|---|---|
| ML | Machine learning |
| DL | Deep learning |
| TL | Transfer Learning |
| MRI | Magnetic resonance imaging |
| CNN | Convolutional neural network |
| SVM | Support vector machine |
| BT | Brain tumors |

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
