# Peer review of "Classifying Brain Tumors on Magnetic Resonance Imaging by Using Convolutional Neural Networks"

_electronics, doi:10.3390/electronics12040955_

Round 1

Reviewer 1 Report

The authors proposed a method for brain tumor classification by using and MIR and CNN. The question is important and the application CNN is okey. The paper is well written. There are already many papers published by CNN for the exact same topic, but not mentioned in the paper. Furthermore, the software is not released and I can't further check their originality. I have two major concerns.

(1) The novelty. Many papers have been published for brain tumor classification by MRI and CNN. Some of them are as follows, but not even discussed.

https://pubmed.ncbi.nlm.nih.gov/35430973/

https://pubmed.ncbi.nlm.nih.gov/33120595/

https://www.sciencedirect.com/science/article/pii/S2665917422000460

https://ieeexplore.ieee.org/document/8934603

https://www.frontiersin.org/articles/10.3389/fnins.2021.679847/full

https://www.mdpi.com/2076-3417/10/6/1999

(2) The software is unreachable. As a method paper, it is unacceptable that there is no public release of the method.

Reviewer 2 Report

The paper analyzes brain tumor recognition from brain MRI images using deep learning methodology. The paper needs to be revised improved to address the provided comments before it could be considered for publication.

Major Comments:

1.       The abstract should mention the name of the dataset used in this study.

2.       The study used some well-known pre-trained deep learning models such as ResNet50, InceptionV3, InceptionResNetV2, Xception, MobileNetV2 and EfficientNetB0. The authors need to explicitly state the novelty of this study.

3.       The literature review is unstructured and difficult to follow. The selection of works of works is unclear. I suggest to introduce structural organization by discussing the works representing various kinds of methodologies such as deep learning, machine learning, hybrid, heuristic. The authors are encouraged to discuss, among others, the excellent works of * Rajinikanth, et al. Y. (2021). Convolutional-neural-network assisted segmentation and svm classification of brain tumor in clinical mri slices. * Khan et al. (2022), Multimodal brain tumor detection and classification using deep saliency map and improved dragonfly optimization algorithm. * Badjie, et al. (2022). A deep transfer learning based architecture for brain tumor classification using MR images.

4.       How the values of hyperparameters given in Table 1 and Table 3 were selected?

5.       The difference between the performance of some models given in Table 4 is very small. Is it statistically significant? Perform statistical analysis using, for example standard deviation or confidence limits calculated from all folds of data, and present the results.

6.       For performance analysis, also present and discuss the confusion matrices of the classification results.

7.       Evaluate the computational complexity of the deep learning models used in this study.

8.       Use some explainability (XAI) methods such as GradCAM to compare between models and explain the results.

9.       Discuss the limitations of the proposed methodology. What is the relevancy of the proposed method for a wider biomedical research field?

Round 2

Reviewer 1 Report

Thank authors for addressing my two major concerns. In details, they have partially responded my first concern, but refused to release their code (the second question). As I mentioned in my first review, the novelty of this research is limited as many similar methods and results already been published. Furthermore, no code release is unacceptable for this paper. Although the authors mentioned this is a research paper, I cann't see what biological discoveries significant enough. Overall, this is a methodology paper.

Author Response

Response

Thank you for your comments and suggestions; for us, they have been very valuable and helped improve the quality of the Manuscript and clarify its contribution to state of the art.

On the other hand, we are sorry that we cannot share the code of the proposed method in this Research Article because we are still in the process of copyright protection and hope to publish it soon in Paper Software. However, in our opinion, this Manuscript is written under the MDPI Instructions for Authors, which specifically for the Materials and Methods Section says, "They should be described with sufficient detail to allow others to replicate and build on published results. New methods and protocols should be described in detail while well-established methods can be briefly described and appropriately cited. Give the name and version of any software used and make clear whether computer code used is available. Include any pre-registration code".

Therefore, as answered in the previous Revision, the main contribution of this Manuscript is in the Preprocessing stage. For this reason, to clarify the proposed method and facilitate its reproducibility, we have added more detail of the Preprocessing stage and updated Figure 1 so that the proposed Preprocessing techniques appear in the sequential order they were coded. The details of this stage can be reviewed in Subsection 2.2. Preprocessing, particularly in lines 227-261.

Finally, the authors would like to thank you for your comments and suggestions, which have contributed to improving the Manuscript's quality and clarifying its contribution to state of the art.

Best wishes, have a wonderful day, and many thanks.

All the authors

Reviewer 2 Report

The authors have addressed all my suggestions and comments and improved the manuscript accordingly. I have no futher comments and recommend the manuscript to be accepted for publication.

Author Response

 Response:

We appreciate all your comments and suggestions made to this Manuscript. We also believe that the quality of the Manuscript has improved considerably. Thank you again for recommending the publication of this Manuscript.

Finally, the authors appreciate your feedback, which enhances the Manuscript and clarifies its contribution to the field.

Greetings. Have a great day, and thanks for your time.

All the authors